

# Uncertainties in shoreline position analysis: the role of run-up and tide in a gentle slope beach

Giorgio Manno[1], Carlo Lo Re[1], and Giuseppe Ciraolo[1]

[1]Department of Civil, Environmental, Aerospace, Materials Engineering – University of Palermo, Viale delle Scienze, Ed. 8, 90128 Palermo (PA)

*Correspondence to:* Carlo Lo Re (carlo.lore@unipa.it)

**Abstract.** In the last decades in the Mediterranean sea, high anthropic pressure from increasing economic and touristic development has affected several coastal areas. Today the erosion phenomena threaten human activities and existing structures, and interdisciplinary studies are needed to better understand actual coastal dynamics. Beach evolution analysis can be conducted using GIS methodologies, such as the well-known Digital Shoreline Analysis System (DSAS), in which error assessment based
on shoreline positioning plays a significant role. In this study, we propose a new approach to estimate the positioning errors due to tide and wave run-up influence. To improve the assessment of the wave run-up uncertainty, we used a spectral numerical model to propagate waves from deep to intermediate water and a Boussinesq-type model for intermediate water up to the swash zone. Tide effects on the uncertainty of shoreline position were evaluated using data collected by a near tide gauge. The proposed methodology was applied to an unprotected, dissipative Sicilian beach far from harbours and subjected to intense
human activities over the last 20 years. The results show wave run-up and tide errors ranging from 0.12 m to 4.5 m and from 1.20 m to 1.39 m, respectively.

## 1 Introduction

Mediterranean beaches are well known for their high environmental, economic, and sociocultural value. In the last few decades, most of these beaches have been subjected to demographic growth from increasing tourism and commercial activities (Cooper
et al., 2009). To support these activities, new defence structures have been built along some beaches, and although these structures have reduced local erosive effects, they have also increased erosion on neighbouring coasts (e.g., Griggs, 2005; Stancheva et al., 2011; Manno et al., 2016). Coastal erosion is a relevant problem that involves both socio economic resources and private properties, and its assessment has long been an issue of international interest involving political decision-makers and researchers (Douglas and Crowell, 2000; Phillips and Jones, 2006; Anfuso et al., 2011; Rangel Buitrago and Anfuso,
2015). Historical beach evolution, erosion, and the retreat/accretion of shoreline has been analysed using aerial and satellite images (e.g., Thieler et al., 2009; Fletcher et al., 2003; Genz et al., 2007; Anfuso et al., 2011; Dolan et al., 1980, 1991). Each remote image is often used to represent a year, and therefore the "shoreline" position identified and digitalized from each image, becomes representative of all shoreline positions in that specific year. The Coastal Engineering Manual (U.S. Army, 2008) defines "shoreline" as the intersection between land and water body, but to choose a suitable proxy that retains the spatial





and time variability (Bush et al., 1999), this boundary much be localized. Among different shoreline proxies (Boak and Turner, 2005), the wet/dry boundary is clearly identified in aerial images by the different colours of sand during the drying process. Because it is more sensitive to run-up fluctuations than astronomical tide variations (Dolan et al., 1980), the wet/dry boundary is a stable shoreline proxy that has been applied by several authors for various applications regarding localization and analysis of shoreline (e.g. Pajak and Leatherman, 2002; Moore, 2000; Moore et al., 2006; Stockdon et al., 2002; Robertson et al., 2004). Thieler et al. (2009) developed a method to assess the beach evolution trend by means of aerial imageries, implemented in a software extension to ESRI ArcGIS© v.9+, the Digital Shoreline Analysis System (DSAS), that can calculate the shoreline rate-of-change statistics starting from multiple historical shoreline positions. This method has the advantage of considering uncertainties due to positioning and measurements errors (Fletcher et al., 2003). The positioning errors are strictly connected to physical phenomena and can affect the analysis precision because the erroneous position of shoreline is assumed as "actual" for the considered year. Uncertainties from tides and wave storms (seasonal variability) are linked to the "exact position" of shoreline during the aerial shooting (Fletcher et al., 2003), whereas measurement uncertainties are linked to errors of image processing and digitizing conducted by technicians who identify and map the shoreline position for several observation years (Fletcher et al., 2003). Several authors (Genz et al., 2007; Rooney et al., 2003; Romine and Fletcher, 2012) used DSAS for studies to evaluate both positioning errors and measurement errors, neglecting the error due to wave run-up and astronomic tide fluctuations. By contrast, other authors (e.g. Virdis et al., 2012; Manca et al., 2013) added to the positioning uncertainty the effects due to wave motion, calculating the run-up by means of the empirical formula of (Hunt, 1959). In this paper, we present an interdisciplinary method that more accurately assesses shoreline positioning error caused by wave run-up and tidal fluctuations in DSAS analysis. Wave run-up was calculated using a numerical model cascade, which includes a wave spectral model and a shallow water propagation model and tide effects were evaluated using the daily variation of astronomic and meteorological tide. With this method, we analysed a dissipative sandy beach of the western coast of Sicily (Italy), an interesting case study because, in the last decades, it has been heavily impacted by human activities. This beach represents a practical case in which accurate identification of the shoreline position with extreme fluctuations is fundamental to forecasting inundation areas or planning effective beach management practices.

## 2 Methodology

Our methodological goal was to better evaluate positioning errors caused by wave run-up and tide for DSAS applications, an ArcGIS extension used to compute the shoreline rate-of-change (Thieler et al., 2009). The latter was evaluated by five different methods to compare the related results. The first method considered the end point rate (EPR), calculated by dividing the shoreline shift by the time elapsed between the oldest and the most recent shoreline position. The second method used the linear regression rate (LRR) of change based on the determination of least-squares regression lines of all the shoreline points of





each transect. The third method used a weighted linear regression (WLR), in which the weight $w$ is a function of the variance of the measurement uncertainty (Thieler et al., 2009): eq. (1):

$$w = 1/e^2 \tag{1}$$

where $e$ is the shoreline uncertainty value. The fourth and fifth methods are based on the analysis of distances rather than

rates. The fourth method considers the "net shoreline movement" (NSM), the distance between the oldest and youngest shoreline position for each transect, and the fifth considers the "shoreline change envelope" (SCE), the distance between the farthest and closest shorelines to the baseline at each transect. To assess the total uncertainty ($\sigma_T$) affecting each shoreline position, we assumed the following relationship (Virdis et al., 2012):

$$\sigma_T = \pm\sqrt{\sigma_d^2 + \sigma_p^2 + \sigma_r^2 + \sigma_{co}^2 + \sigma_{wr}^2 + \sigma_{td}^2} \tag{2}$$

where the uncertainty $\sigma_i$ is the standard deviation of the $i$-type error; $\sigma_d$ is the digitizing error determined by digitizing several times the same feature on the image; $\sigma_r$ is the orthorectification error, considered as the root mean square error (RMSE) for photogrammetric blocks; $\sigma_{co}$ is the image coregistration error arising from the RMSE of misalignment between single pixels from the set of images obtained by the rectification; $\sigma_p$ is the pixel error assumed equal to the pixel size; and $\sigma_{wr}$ and $\sigma_{td}$ are, respectively, the wave run-up and the tide errors estimated in this study (discussed later). Note that the first four errors

relate to intrinsic characteristics of the used images, how they were taken and how they were processed, whereas the last two relate to specific geomorphologic, mareographic, and wave characteristics of the beach examined. Variables $\sigma_{td}$ and $\sigma_{wr}$ represent position errors that may result in noticeably higher values than the others; therefore special care is required during their evaluation, which is the focus of this study.

To improve the evaluation of the wave run-up and tide uncertainty ($\sigma_{wr}$, $\sigma_{td}$) with respect to the use of empirical formulas

found in the technical literature (e.g. Virdis et al., 2012; Manca et al., 2013), we applied various mathematical models. An hydraulic study, conducted on the basis of a geomorphologic study, determined the effects of wave motion and tide fluctuation on the shoreline position. To this aim, offshore wave parameters were used to simulate wave propagation from deep water to run-up on the beach, whereas a tide-gauge dataset was used for analysis of tide fluctuation. The whole mathematical process for the run-up calculation can be summarized by the following steps: a) select offshore buoy dataset collection; b) propagate waves

from deep to intermediate water by means of a wave spectral model; c) generate random waves from a JONSWAP spectrum; d) propagate waves from intermediate water up to the swash zone with a Boussinesq-type model; and e) conduct run-up analysis. This mathematical process is validated using in field measurements as described in the Section 4. The propagation of the offshore wave characteristics to shallow water was carried out by the well-known SWAN spectral propagation model (Booij et al., 1999; Holthuijsen et al., 1993; Ris et al., 1999). The SWAN results obtained for the 5 m bathymetric line were

then used as input for the Boussinesq-type model by Lo Re et al. (2012b) which, coupled with a specific Lagrangian model for shoreline movement, allowed simulation of wave swash and run-up. The wave run-up error, $\sigma_{wr}$, was finally estimated





by analysing the resulting shoreline movement over time. Note that the offshore wave parameters were the only source in the propagation model SWAN, and wind, bottom friction, and white-capping were not considered. For wave propagation by SWAN, a 2D unstructured grid following the Delaunay rule was implemented, constructed in accordance with Monteforte et al. (2015) using a density function in which the triangle sizes depended on local water depth and wavelength. The node elevation

was calculated by a linear interpolation of bathymetric data from nautical charts. The Lagrangian model used for shoreline movement discriminates between wet and dry regions to simulate run-up and run-down along surveyed sections. For the $i$th section, the standard deviation of the horizontal shoreline movement over time was calculated by:

$$\sigma_{w,i} = \frac{S_{wr,i}}{\tan \alpha_i} \tag{3}$$

where $S_{wr,i}$ is the standard deviation of the vertical shoreline fluctuation computed by the model, and $\tan \alpha_i$ is the section

slope. Finally, the wave run-up error for the whole beach, $\sigma_{wr}$, was estimated by:

$$\sigma_{wr} = \sqrt{\frac{\sigma_{wr,1}^2 + \sigma_{wr,2}^2 + ... + \sigma_{wr,n}^2}{(n-1)}} \tag{4}$$

The tide uncertainty $\sigma_{td}$ was assessed by processing the tide measurements recorded by a mareographic station. For each year with measures, the standard deviation of the tide measurements, $S_{td}$, was first computed, and the standard deviations of the horizontal tidal fluctuations of the same sections used for run-up assess were then evaluated using an equation formally

identical to Eq. (3). The tide uncertainty for the whole beach, $\sigma_{td}$, was finally assessed with the same equation used for the run-up error (Eq. (4)).

## 3   The case study: *Marsala* beach

The case study of the dissipative beach (Fig. 1), *Lido Signorino*, extends in a north–south direction for about 3.5 km between Cape Torre Tunna and Cape Torre Sibilliana. Its slope ranges between 1.5° and 10.8°, and the direction of beach exposure (Fig.

1) is about 140°, between NW and S-SW. The *Egadi* Islands, in particular *Favignana* Island, shield the beach in the 320° N direction. The geographical fetch is limited from the west by the Spanish coast, from the south by the African coast, and from the north-west by the Sardinian coast.

The buoy belongs to the Italian Wave Buoy Network (RON), and the rose, obtained by processing available data recorded from July 1989 to June 2012, shows that the most intense and most numerous waves come from around 270° N, 290° N

and 292.5° N. The beach is made of fine carbonatic sand (Holocene) with sub-smoothed lithic and fossil shells grains. The granulometric analysis by Manno et al. (2011) indicates $D_{10} = 0.20$ mm, $D_{30} = 0.33$ mm, $D_{60} = 0.55$ mm, and a uniformity coefficient of 2.75 mm. The granulometric fractions are 0.4% silt, 0.6% clay, and 99% sand. The sediment has effective porosity of about 26% and high permeability ranging between $10^{-2}$ and $5 \cdot 10^{-3}$ cm/s.

The beach suffers from intense anthropogenic use, especially housing built close to the shoreline (Fig. 2), which has caused

progressive destruction of the dunes and the natural sand supply. In the first 50 years of the 20th century, the dunes were




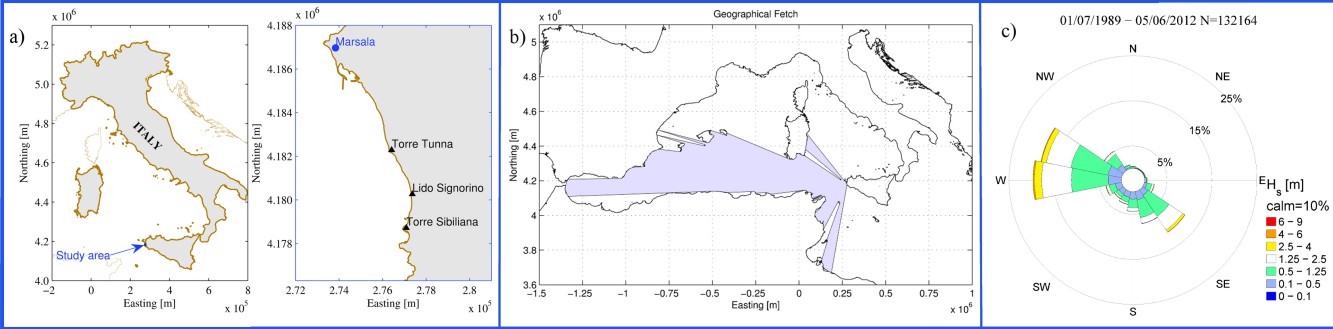

**Figure 1.** a) Map of Italy (left panel) and of the study area (right panel), showing the locations of Cape *Torre Tunna*, Cape *Torre Sibilliana*, *Lido Signorino* beach and *Marsala* City (ED50-UTM33N Coordinate Reference System); b) Direction of exposure and geographical fetch of *Lido Signorino* beach. c) Wave rose at the *Mazara del Vallo* buoy, relating to the period 01/07/1989 to 05/06/2012.

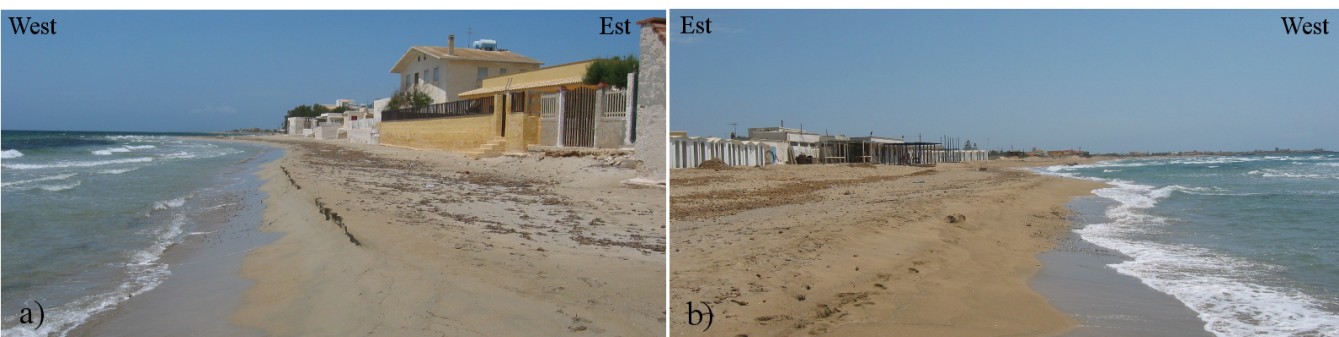

**Figure 2.** Anthropic pressure in the studied beach: a) central-northern beach, where buildings are at about 4-5 m from the shoreline and are reached by waves during sea-storms; b) central-southern beach, where buildings are noticeably farer from the shoreline.

uniform from north to south and about 5 m high, whereas today they are discontinuous, about 2.5 m high and mainly located in the less developed southern area.

## 4   Validation of run-up assessment by means of field measurements: a *Marsala* beach

In order to validate the whole mathematical process used for run-up assessment, field measurements were performed (Lo Re et al., 2012a). The wave run-up on sandy beaches can be measured in several ways depending on the general aim and on the amount of details required. Records of the shoreline positions can be obtained by resistance run-up gauges or by video-cameras as applied by Holman and Sallenger (1985).

The technique adopted in present paper is based on a high frequency monitoring video system. Such kind of technique allows the acquisition of several images by means of a digital video camera. The choice of the position of the camera was a fundamental task because the camera has to shoot the whole studied area but at a little distance, in order to obtain the maximum





level of detail from the recorded images. In particular, positions of the swash were measured on a transect across the beach, normal to the shore ( Fig. 3). For this transect a line was built using stakes at 0.5 m intervals. The first stake was a piezometer and it was next to the beach berm. The second stake of the line was placed at a distance of 5 m from the piezometer. The line stakes on the beach profile was georeferenced using control points from a previous topographic survey. The video camera was

placed at a distance of 10 m from the line of stakes (orthogonally), and it was used to record 240 minutes in continuous. The shot videos were digitized in order to extract the wave run-up of each wave. When a wave reached a stake the data was recorded. The horizontal run-up distance were calculated starting from SWL obtained from water level inside the piezometer. Finally the corresponding run-up value was estimated by considering the beach profile. Each run-up measurement ($R$) was recorded in time windows of thirty minutes (eight windows in total) accordingly to Nielsen and Hanslow (1991). For all recorded data,

the Rayleigh distribution was fitted by using the least squares method. The application of the Rayleigh distribution to our data allowed to estimate the 2% run-up ($R_{2\%}$).

The expression of the Raylegh cumulative distribution function is reported in the following:

$$F(R) = 1 - exp\left\{-\frac{(R - R_{100})^2}{L_{zwm}^2}\right\} \tag{5}$$

in which $R_{100}$ is the value transgressed by 100% of the waves, i.e. the lower limit of the distribution, and $L_{zwm}$ is the vertical

scale of the distribution, i.e. the shape parameter.

Moreover to perform such a validation wave parameters from buoy of *Mazara del Vallo* were used. In particular: 1) significant wave heights, $H_s$ [m]; 2) peak period $T_p$ [s] and 3) mean wave direction $D_m$ [°N]. The extraction time period goes from 11:30 to 15:30 of 29 march 2011. The waves shown in Tab. 1 correspond to the sea states recorded by the buoy half-hourly.

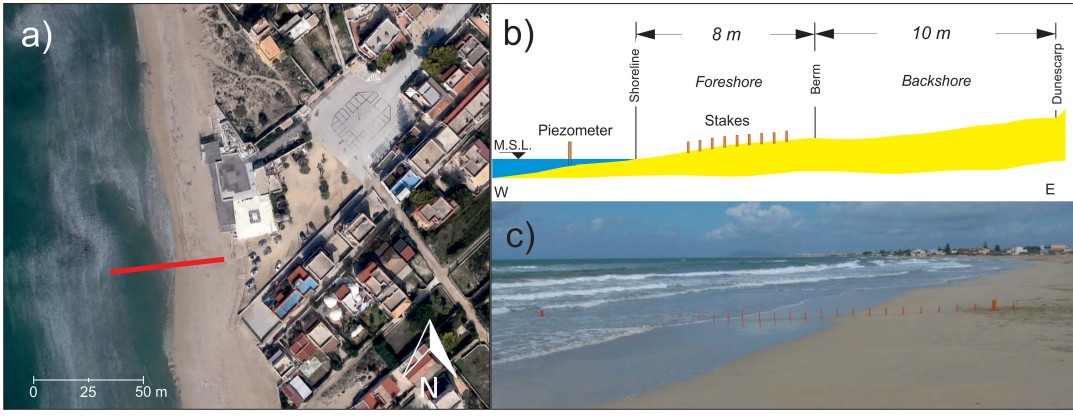

**Figure 3.** Plan view (a) cross section sketch (b) and beach profile (c) of the reference transect n°45 (see following Section) for the run-up measurements at *Lido Signorino* beach



The obtained wave run-up are reported in Tab. 1. Such a table also shows, for each time window, the results obtained with the empirical formula by Nielsen and Hanslow (1991). The $R_{2\%}$ run-up determined by means of the Rayleigh distribution of field measurements is also shown.

**Table 1.** $R_{2\%}$ run-up comparison between field measurements, numerical model (Lo Re et al., 2012b) and empirical formula (Nielsen and Hanslow, 1991).

| | Offshore | | Measured | Boussinesq | | Nielsen and Hanslow (1991) | |
|---|---|---|---|---|---|---|---|
| Wave# | $H_s$ [m] | $T_p$ [s] | $R_{2\%}[m]$ | $R_{2\%}[m]$ | error [%] | $R_{2\%}[m]$ | error [%] |
| 1 | 1.13 | 6.13 | 0.89 | 0.90 | 0.46 | 0.57 | 35.95 |
| 2 | 1.05 | 7.27 | 0.93 | 0.91 | 2.86 | 0.80 | 14.65 |
| 3 | 1.07 | 5.94 | 0.86 | 0.82 | 5.26 | 0.62 | 28.58 |
| 4 | 1.04 | 7.18 | 0.84 | 0.84 | 1.08 | 0.84 | 1.06 |
| 5 | 1.03 | 7.25 | 0.89 | 0.87 | 2.78 | 0.80 | 10.39 |
| 6 | 1.10 | 7.26 | 0.89 | 0.89 | 0.28 | 0.83 | 6.96 |
| 7 | 0.99 | 6.93 | 0.91 | 0.82 | 8.95 | 0.68 | 24.55 |
| 8 | 1.05 | 6.33 | 0.91 | 0.91 | 0.83 | 0.70 | 22.76 |
| | | | | **mean** | **2.81** | | **18.11** |

The analysis of the 2% run-up ($R_{2\%}$) highlights that both methods give acceptable results. In particular, the numerical model has an average percentage error of 2.81% and the empirical formula gives an average percentage error of 18.11%. The numerical Boussinesq model gives overall result closer to the field measurements and for this reason it was chosen for simulations of wave run-up in this study.

## 5 Method application and results

Five orthorectified aerial images taken during 1994–2007 were used to assess time variations of the shoreline position during the period (Table 2). Each image was georeferenced (WGS84 - UTM 33N) by 6-10 evenly spaced control points. For each observation year, these images were used to form a photo-mosaic covering the whole coast studied. The shoreline relating to a photo-mosaic was traced and digitized manually using the wet/dry proxy, as suggested by Virdis et al. (2012) for dissipative beaches of the Mediterranean Sea.

For each of the five aerial surveys, an offshore wave condition was obtained by processing the measurements of the *Mazara del Vallo* buoy (Fig. 4) taken during the time period of the survey (Table 4). In particular, 3 hourly wave parameters were used for a total of 40 sea states for 5 days (each day has its specific beach profile, wave and tide). Every single sea state (see Table 4) was then propagated throughout the numerical domain by SWAN in stationary mode (Fig. 4). At the 5 m bathymetric line, the wave spectrum output of SWAN was used to generate a wave time-series (Table 4), which in turn was used as input to the



**Table 2.** Characteristics of the aerial images used for the analysis of the case study

| Images name | Data | Spatial resolution (m) | Film | Altitude fly [m] |
|---|---|---|---|---|
| Volo Italia 1994 | 07 June 1994 | 1 | Black and white | 11 500 |
| Volo Italia IT2000 | 13 May 1999 | 1 | Colour | 6 000 |
| Acquater | 15 October 2000 | 4 | Colour | 3 000 |
| Volo Italia IT 2006 | 29 September 2006 | 0.5 | Colour | 3 000 |
| ECW 2007 | 16 September 2007 | 0.5 | Colour | 3 000 |

Boussinesq model to assess the wave run-up in 26 sections. Table 3 shows the average slope for each day analysed. Therefore, 1 040 (26 sections × 40 sea state × 5 days) near-shore simulations were conducted for each offshore wave.

The considered sections (Fig. 4) are distinguished from the transects (discussed later) by an S preceding the number. In addition to the 26 sections, 68 transects orthogonal to the present shoreline (Fig. 4) were generated (with DSAS application) at about 50 m from one another to better analyse the shoreline changes and the related erosion/accretion rates between the transects themselves.

**Table 3.** Average beach slope in the analysed days.

| Section# | Slope | $\alpha$ [°] | Section# | Slope | $\alpha$ [°] |
|---|---|---|---|---|---|
| 1 | 11.55 | 4.95 | 14 | 8.64 | 6.61 |
| 2 | 10.55 | 5.42 | 15 | 8.48 | 6.72 |
| 3 | 10.07 | 5.67 | 16 | 8.14 | 7.01 |
| 4 | 11.44 | 4.99 | 17 | 12.38 | 4.62 |
| 5 | 10.80 | 5.29 | 18 | 11.12 | 5.14 |
| 6 | 11.53 | 4.96 | 19 | 21.93 | 2.61 |
| 7 | 13.62 | 4.20 | 20 | 15.48 | 3.70 |
| 8 | 11.79 | 4.85 | 21 | 10.98 | 5.21 |
| 9 | 14.64 | 3.91 | 22 | 23.47 | 2.44 |
| 10 | 10.79 | 5.30 | 23 | 18.15 | 3.15 |
| 11 | 15.80 | 3.62 | 24 | 17.99 | 3.18 |
| 12 | 7.93 | 7.19 | 25 | 15.67 | 3.65 |
| 13 | 9.22 | 6.19 | 26 | 15.70 | 3.64 |



Table 4: Data and results of the simulations relating to the five aerial surveys: day considered for wave data processing, offshore wave data used as input for SWAN, and SWAN output used as Boussinesq-model input.

| Survey Date | Offshore wave data → SWAN-input | | | SWAN out → Boussinesq in | |
|---|---|---|---|---|---|
| [YYYY MM DD HH] | $H_s$ [m] | $T_p(H_s)$ [s] | $Dir$ [°] | $H_s$ [m] | $T_p$ [s] |
| 1994 06 07 00 | 1.00 | 6.30 | 279 | 0.82 | 6.20 |
| 1994 06 07 03 | 0.90 | 5.90 | 276 | 0.75 | 6.00 |
| 1994 06 07 06 | 0.90 | 6.70 | 278 | 0.72 | 6.48 |
| 1994 06 07 09 | 0.80 | 6.30 | 280 | 0.68 | 6.23 |
| 1994 06 07 12 | 0.90 | 6.30 | 282 | 0.72 | 6.20 |
| 1994 06 07 15 | 0.90 | 4.50 | 289 | 0.80 | 4.49 |
| 1994 06 07 18 | 0.70 | 4.20 | 287 | 0.61 | 4.13 |
| 1994 06 07 21 | 0.60 | 5.60 | 281 | 0.50 | 5.51 |
| 1999 05 13 00 | 0.20 | 2.30 | 306 | 0.20 | 2.29 |
| 1999 05 13 03 | 0.20 | 4.80 | 268 | 0.17 | 4.71 |
| 1999 05 13 06 | 0.20 | 5.00 | 270 | 0.17 | 4.86 |
| 1999 05 13 09 | 0.20 | 4.00 | 324 | 0.19 | 3.95 |
| 1999 05 13 12 | 0.26 | 4.09 | 318 | 0.24 | 4.07 |
| 1999 05 13 15 | 0.32 | 4.18 | 313 | 0.29 | 4.13 |
| 1999 05 13 18 | 0.38 | 4.27 | 307 | 0.34 | 4.18 |
| 1999 05 13 21 | 0.44 | 4.35 | 302 | 0.39 | 4.25 |
| 2000 10 15 00 | 2.74 | 7.10 | 110 | 2.22 | 7.11 |
| 2000 10 15 03 | 2.02 | 6.70 | 154 | 1.91 | 6.67 |
| 2000 10 15 06 | 1.50 | 6.70 | 146 | 1.63 | 6.67 |
| 2000 10 15 09 | 1.30 | 6.70 | 177 | 1.28 | 6.67 |
| 2000 10 15 12 | 1.27 | 6.70 | 183 | 1.21 | 6.67 |
| 2000 10 15 15 | 1.24 | 6.70 | 189 | 1.14 | 6.67 |
| 2000 10 15 18 | 1.21 | 6.70 | 196 | 1.08 | 6.67 |
| 2000 10 15 21 | 1.18 | 6.70 | 202 | 1.05 | 6.67 |
| 2006 09 29 00 | 0.65 | 6.70 | 277 | 0.53 | 6.65 |
| 2006 09 29 03 | 0.62 | 5.70 | 258 | 0.49 | 5.68 |
| 2006 09 29 06 | 0.62 | 5.60 | 268 | 0.54 | 5.56 |
| 2006 09 29 09 | 0.57 | 4.76 | 276 | 0.52 | 4.77 |
| 2006 09 29 12 | 0.52 | 5.13 | 295 | 0.44 | 5.05 |





Table 4: Data and results of the simulations relating to the five aerial surveys: day considered for wave data processing, offshore wave data used as input for SWAN, and SWAN output used as Boussinesq-model input.

| Survey Date | Offshore wave data → SWAN-input | | | SWAN out → Boussinesq in | |
|---|---|---|---|---|---|
| [YYYY MM DD HH] | $H_s$ [m] | $T_p(H_s)$ [s] | $Dir$ [°] | $H_s$ [m] | $T_p$ [s] |
| 2006 09 29 15 | 0.45 | 4.50 | 278 | 0.37 | 4.49 |
| 2006 09 29 18 | 0.37 | 4.50 | 264 | 0.32 | 4.47 |
| 2006 09 29 21 | 0.32 | 4.55 | 261 | 0.29 | 4.50 |
| 2007 09 16 00 | 0.43 | 5.90 | 202 | 0.38 | 5.90 |
| 2007 09 16 03 | 0.37 | 6.70 | 215 | 0.33 | 6.63 |
| 2007 09 16 06 | 0.36 | 4.76 | 226 | 0.32 | 6.62 |
| 2007 09 16 09 | 0.35 | 5.41 | 232 | 0.31 | 6.49 |
| 2007 09 16 12 | 0.30 | 4.44 | 222 | 0.27 | 4.42 |
| 2007 09 16 15 | 0.34 | 4.76 | 187 | 0.31 | 4.63 |
| 2007 09 16 18 | 0.32 | 2.17 | 181 | 0.31 | 4.64 |
| 2007 09 16 21 | 0.40 | 3.51 | 190 | 0.39 | 3.45 |

*Concluded*

The Boussinesq wave propagation on 26 sections produced a number of run-up values that were then processed, obtaining the 2% wave run-up ($R2_{\%,i}$) and the run-up standard deviation $\sigma_{wr,i}$ (i = 1, 2, ..., 26) relating to each survey day (Fig. 5). Comparison between the run-up values and the standard deviation indicates, for each year, consistency between the $R_{2\%}$ and $\sigma_{wr,i}$ trends (Fig. 5). The wave run-up errors $\sigma_{wr}$ (Eq. (4)) were summarized Table 5, together with the other uncertainties.

5     For each day with measurements, the standard deviation of the tide measurements ($S_{td}$) was first computed and then the standard deviations of the horizontal tidal fluctuations of the 26 sections were evaluated by Eq. (4). As is well known tide fluctuation measurements include the meteorological effects.

**Table 5.** Standard deviations $S_{td}$ of the tide fluctuations gauged at near station and the related tide uncertainties $\sigma_{td}$ for *Lido Signorino* beach.

| Date | 07 June 1994 | 13 May 1999 | 15 October 2000 | 29 September 2006 | 16 September 2007 |
|---|---|---|---|---|---|
| $S$ [m] | 0.09 | 0.10 | 0.10 | 0.09 | 0.09 |
| $\sigma_{td}$ [m] | 1.31 | 1.37 | 1.39 | 1.28 | 1.20 |

Based on the uncertainties (Table 6), the five shoreline rate-of-change indices mentioned earlier were evaluated using DSAS for each of the 68 transects. Indices WLR, EPR and LRR (Fig. 6) and indices NSM and SCE (Fig. 7) were plotted, with positive 10  index values indicating shoreline accretion and negative values indicating recession.





**Figure 4.** a) The numerical domain Monteforte et al. (2015). The blue bold line in the left panel represents the domain boundary along which the offshore wave conditions were imposed; The left panel represents the bathymetry of studied area) Location of the 26 sections chosen for the run-up analysis; each section is identified by a number preceded by S that distinguishes them from the transects, identified by a number only.





**Figure 5.** a) $R_{2\%}$ of the wave run-up output of Boussinesq model; b) standard deviation of the shoreline movement due to wave swash.

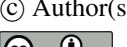



**Table 6.** Uncertainties determined for the shoreline position [m].

| Date | 07 June 1994 | 13 May 1999 | 15 October 2000 | 29 September 2006 | 16 September 2007 |
|---|---|---|---|---|---|
| $\sigma_d$ | 4.10 | 4.10 | 11.00 | 0.60 | 2.00 |
| $\sigma_p$ | 1.00 | 1.00 | 4.00 | 0.50 | 0.50 |
| $\sigma_r$ | 1.00 | 1.00 | 4.00 | 0.50 | 0.50 |
| $\sigma_{co}$ | 1.00 | 1.00 | 4.00 | 0.00 | 0.00 |
| $\sigma_{wr}$ | 2.41 | 0.01 | 4.50 | 1.36 | 0.12 |
| $\sigma_{td}$ | 1.31 | 1.37 | 1.39 | 1.28 | 1.12 |
| $\sigma_T(eq.(6))$ | **5.23** | **4.66** | **13.83** | **2.09** | **2.40** |

The variations among WLR, EPR, and LRR (Fig. 6a) are consistent, exhibiting generally similar trends both on accretion and on recession. Several accretion zones are clearly distinguishable in transects 12-13, 33-40, 47, and 65-67. The accretion rate is about 0.6 m/year for transect 47, 0.51 m/year for transects 12-13, and 1.0 m/year in transects 33-40, 47, and 65-67. By contrast, a recession is evident for transects 56-64, with a rate about –2.5 m/year, as well as for transects 48-50 of –1.6 m/year. For transects 20-22 and 40-43, lower recession rates of –1.18 and –1.28 m/year, respectively, are observed. In contrast to the other indices, NSM and SCE often present opposing trends (Fig. 6b), in which a relative maximum of one may correspond to a relative minimum of the other. This pattern depends on the different definitions of the two indices, but disparate conclusions can be drawn if one or the other criterion is adopted. Indeed SCE represents the total change in shoreline movement for all available shoreline positions and is not related to their dates. Fig 6b shows that NSM trend on the whole is consistent with the other indices trend (Fig 6a), except for transects 36-37 and 44-45. The lowest shoreline indices values (stable areas) were generally localized in transects from 3 to 9 (8 cusp), from 22 to 34 (24 cusp) and from 50 to 55.

In particular, in transect 11 a deposition cusp was detected. Transects 51 and 55 proved to be affected by even lower shoreline movements, despite considerable anthropogenic disturbances of the dunes, this for all the indexes showed in Fig 6a,b. Unlike the SCE index, the NSM trend (Fig. 6b) showed a reliable average equilibrium between beach accretion and recession, although in transects 56-64 a noticeable general recession occurs, in accordance with EPR, WLR, and LRR results (Fig. 6a).

Finally, we compared the NSM indices relating to each period between two consecutive surveys to highlight the shoreline evolution at each transect along the whole period 1994–2007 (Fig. 7), showing that accretion periods alternated with recession periods at each transect. In a few transects (e.g., transects 11 and 19) NSM was effectively unchanged, whereas in many others it changed noticeably from one period to another. Moreover, during 2006–2007, accretion prevailed over recession along the entire beach, whereas recession was prevalent during 2000–2006, and during 1994–1999 and 1999–2000, accretion and recession basically balanced each other. The different behaviours (higher or lower accretion or recession) of different beach stretches observed in a given period followed the specific beach conformation of the stretches as well as the presence of *Posidonia oceanica* leaves deposits. Note that all the indices considered detected higher shoreline changes in transects 58-68, where vast deposits of *Posidonia oceanica* leaves were present.







**Figure 6.** a) comparison among the change-rate of the shoreline position at each transect during the studied period, expressed by: End Point Rate (EPR), Linear Regression Rate (LRR), and Weighted Linear Regression rate (WLR); b) Comparison between the Net Shoreline Movement (NSM) and Shoreline Envelope of Changes (SCE) during the studied period.



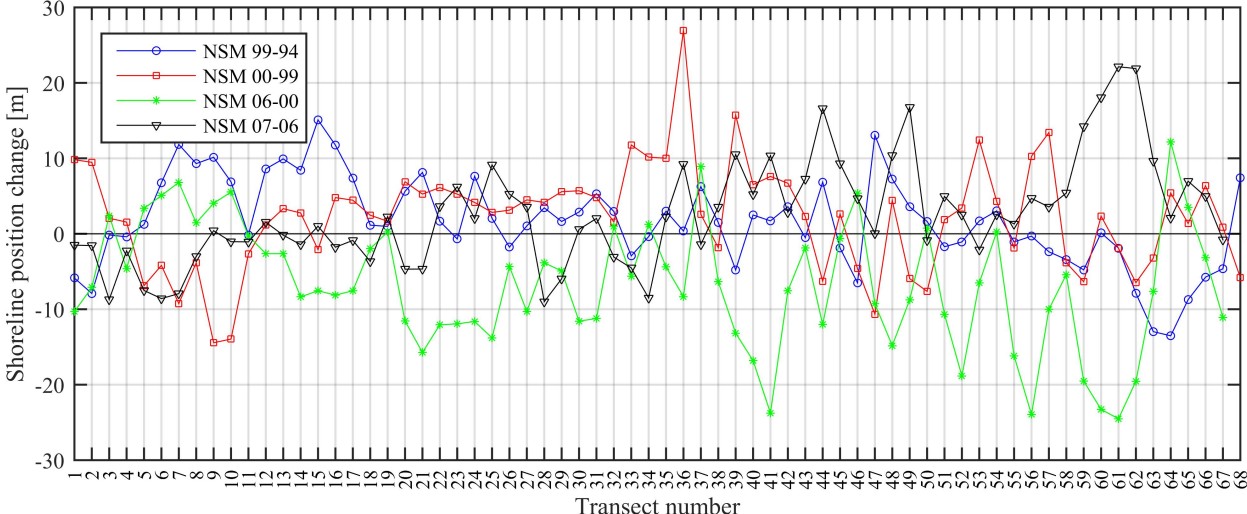

**Figure 7.** Comparison between the Net Shoreline Movement (NSM) relating to each period between two consecutive surveys.

## 6 Conclusions

To analyse beaches by aerial images utilizing DSAS or similar applications, technicians usually neglect positioning uncertainties or assess them by means of empirical formulas without the use of hydraulic models. In this study, we adopted a new approach that assesses positioning uncertainties by analysing wave motion and tide effects.

The hydraulic models used in our methodological approach consist of a nearshore model (SWAN) and a more accurate dispersive model (Boussinesq) that provides a more accurate description of the hydrodynamic in the nearshore area, a fundamental step to estimate the oscillation of the shoreline.The proposed method can be used in all types of coastal areas (steep beaches, gentle slope beach etc.) and can reproduce the hydrodynamic of a large area, not just the hydrodynamic of one point. Moreover, the models can reproduce most all wave propagation effects (diffraction, refraction, reflection, shoaling, breaking,

etc.). The diachronic analysis on *Lido Signorino* shows a low shoreline variability, except for the most southern coastal stretch that has a high variability due to anthropic and natural causes.

    The methodology adopted here provides high accuracy in wave run-up calculation, resulting in a more accurate $\sigma_{wr}$ error estimate as highlighted from comparison of present model with in fields run-up measurements(see Section 4).

    Using an application like DSAS that neglects or underestimates the $\sigma_{wr}$ error may prevent determining if a beach is in

retreatment or advancement, and a retreatment rate close to the total uncertainty would not be constructive. Furthermore, method accuracy is valuable in beach monitoring and management, especially when more sustainable methods are needed to sustain coastal resources. An integrated management of the coasts must be interdisciplinary and consider the dynamic process of the beaches, mainly when the beaches are largely urbanized and anthropized.





*Acknowledgements.* We thank Gino Dardanelli for assistance with topographic survey of the studied area, and for sharing with us his expertise in topographic measurements. We would like to express our gratitude to Giovanni Battista Ferreri for sharing his pearls of wisdom with us during the course of this research. We thank Massimiliano Monteforte for his comments.



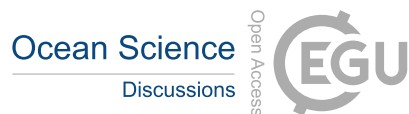

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
