# Peer review of "Uncertainties in shoreline position analysis: the role of run-up and tide in a gentle slope beach"

_Ocean Science, 2017_

## Short Comment (SC1) · 10 May 2017

I find the proposed work interesting, it seems to have a certain degree of originality, and it might be appropriate for a journal publication.

There are however some improvements and corrections that can be done. Some most significant observations are given next:

I would recommend avoiding the first person in the text in the favor of a more impersonal mode.

A separate section of Discussions will provide probably a more comprehensive picture of the results.

[Figure]

Probably the most important issue relates the fact that the validation of the proposed approach is not sufficiently well explained. This includes the reliability of each numerical model considered (the spectral wave model and the dispersive Boussinesq model).

Finally, I was not able to see any keyword, are the keywords not required in this journal?

---

## Short Comment (SC2) · 15 May 2017

We thank Prof. E. Rusu for his thorough and critical evaluation of our manuscript, and for his suggestions, which we will largely take into account in the next stage of revision.

Best Regards

———————————————————

---

## Author Response (AR1)

We wish to thank **Prof. Eugen Rusu** for the comments that helped us to improve the manuscript.

COMMENTS AND RESPONSES

Question / Adjustments
*I would recommend avoiding the first person in the text in the favour of a more impersonal mode.*

Response: Such kind of sentences are modified in the revised manuscript.
——————-
Question / Adjustments
*A separate section of Discussions will provide probably a more comprehensive picture of results.*

Response: We add to revised manuscript a new section entitled "Discussions".
——————-
Question / Adjustments
*Probably the most important issue relates the fact that the validation of the proposed approach is not sufficiently well explained. This includes the reliability of each numerical model considered (the spectral wave model and the dispersive Boussinesq model).*

Response: The validation of the used methodology was better explained in the revised paper. In particular a better detailed description of the Boussinesq model and more information about SWAN model reliability were included in the amended paper.
——————-
Question / Adjustments
*Finally, I was not able to see any keyword, are the keywords not required in this journal?*

Response: The keywords are not required in OC journal.

[Figure]

We wish to thank reviewer#2 for the comments that helped us to improve the manuscript

COMMENTS AND RESPONSES

Question/ Adjustments *Specific comments Pag. 4, I noticed this affirmation: Note that the offshore wave parameters were the only source in the propagation model SWAN, and wind, bottom friction, and white-capping were not considered which are the implications of this? Could you briefly comment it?*

Response: The original manuscript was modified following the reviewer suggestions. The relatively small area of numerical SWAN domain means that wave growth is minimal and thus to simplify the analysis the wind source term is not included in the our analyses. The white-capping was not included in the calculation because generally, it is not recommended to include this source term when there is no wind input. The bottom friction may play an important role in shallow water studies but it is a topic outside the subject of the present work, which is focused mainly on modelling offshore and/or intermediate wave conditions.

———————————-

Question / Adjustments
*Could you please give further information about the Boussinesq model you used? What about the approximation order? Last, I would mind to ask further information about the Lagrangian model used for the shoreline boundary conditions.*

Response: The reviewer is referred to the work of Musumeci et al.(2005) for the analytical details of the derivation of the governing equation of the Boussinesq model. In particular the governing Boussinesq equations has no assumptions about the order of magnitude of the nonlinear parameter $\delta = a_0/h_0$, and the resulting model is fully nonlinear to terms of O($\mu^2$). Where $a_0$ is the offshore wave amplitude, $h_0$ is the offshore water depth, $\mu = k_0 \cdot h_0$ is the dispersive parameter and $k_0$ is the wave number offshore. Dealing with the shoreline motion is a critical issue in numerical models because is necessary to discriminate between the wet region of the computational domain, where calculations of the governing equations are required, and dry region, where no wave motion is defined. In the present paper we used an approach which describes the physics of the swash zone hydrodynamics, by solving the equations of the shoreline motion. Consequently we used specific physically-based equations to calculate the velocity of the shoreline and the shoreline position, which can be solved once the velocities in the remaining (wet) domain are known. Fundamentally, we followed a Lagrangian approach similar to the one presented by Prasad and Svendsen (2003). The shoreline equations are now added in the reviewed paper.
* * *
Question / Adjustments

*Pag. 4, lines 26-27, Not clear, please - if possible - give values such as Median, $D_{50}$ and sorting. ...I guess they are important parameters determining infiltration.*

Response: The $D_{50}$ was added in the amended paper and consequently redundant parameters were removed from paper. The sentence (lines 26-27) was simplified and clarified.
* * *
Question / Adjustments

*13, line 11...which is the importance/implications of the presence of beach cusps? Pag. 13, lines 12-13...which are implications? Even this is NOT strictly related to the topic presented in your paper, I would mind you briefly comment on this topic and implications to your study: I guess cusps' presence can give rise to erroneous results, please consider the paper: Giorgio Anfuso Dan Bowman Chiara Danese Enzo Pranzini (2016) Transect based analysis versus area based analysis to quantify shoreline displacement: spatial resolution issues. Environ Monit Assess, 188:568?*

Response: The beach studied has not a rhythmic morphology. We agree with the reviewer#2 that in the case of large presents of beach cusps they can influence the attended results. Therefore the paper was modified taking into account the remarks made by the reviwer#2.
* * *
Question / Adjustments

*Pag. 15, line 14, change retreatment or advancement for erosion and accretion. And change this and a retreatment rate close to the total uncertainty would not be constructive for: ...a retreat rate close to the total uncertainty would not be acceptable.*

Response: The sentence was modified in the amended paper as suggested by reviewer#2.
* * *
Question / Adjustments

*Technical corrections The quality of English is generally quite good but I have to propose small corrections. Pag. 2, line 1, maybe is "must" and not "much". Pag. 2, lines 14 and 15, I suggest ..."used DSAS to evaluate both positioning errors..." Pag. 2, line 17, Hunt is not in parenthesis, I guess. Pag. 2, line 20, I suggest:..."water propagation model. Tide effects..." Pag. 3, line 15, I suggest:..." four errors are related to " Pag. 4, line 29, I suggest..."especially houses emplaced too close to..." Pag. 4, line 30, I C2 OSD Interactive comment Printer-friendly version Discussion paper suggest:..."destruction of dunes and their associated natural supply..." Pag. 5, Figure 1: I suggest:..."Mazzara del Vallo buoy, related to..." Pag. 5, line 6, I suggest, but not sure..."details of expected results"... Pag. 5, line 7, I suggest to say:"...by Holman and Sallenger (1985), and this is the case of this paper. Based on a high..." Pag. 6, line 16, ..."from the buoy.. ." Pag. 7, line 6, ...to the field measurements and, for this reason, ..." Pag. 7, line 10, I suggest: Five orthorectified aerial images were used to assess time variations of the shoreline position during the 1994–2007 time span (Table 2).*

Response: All the suggested correction were made in the reviewer paper.
* * *
Question/ Adjustments *Pag. 7, line 10, say: ..ground control points... Pag. 7, lines 14-15, this is not clear: For each of the five aerial surveys, an offshore wave condition was obtained by processing the measurements of the Mazara del Vallo buoy (Fig. 4) taken during the time period of the survey (Table 4). I suggest: In order to reconstruct waves conditions at the day the aerial photos were made, data recorded from the Mazzara del Vallo buoy were analysed. Pag. 10, is table 5? Pag. 11, Fig. 4, the letter "b" is missing. Pag. 15, line 13, ...I suggest..."line estimation"...and:..."in situ run-up..."*

Response: This correction was made as suggested by the reviewer#2.
* * *
**List of all relevant changes**

- From page 3 line 27 to page 4 line 18: Added details on Boussinesq numerical model including the governing equations of shoreline boundary condition as suggested by Reviewer#2;
- Page 4 from line 24 to line 27: the SWAN assumptions for source terms were clarified as suggested by Reviewer#1;
- Page 9 from line 1 to 7: The sentences were corrected to better clarify the method adopted;
- Was added a new section at page 14: "Discussion*: comparison of shoreline rates of change*".
- Two papers was added In the references list (page 18 line 3; page 19 line 19).

Summary
18/07/2017 10:28:48

Differences exist between documents.

**New Document:**
sh2017
19 pages (14.79 MB)
18/07/2017 10:28:37
Used to display results.

**Old Document:**
os-2017-18_original
18 pages (14.78 MB)
18/07/2017 10:28:37

Get started: first change is on page 1.

No pages were deleted

**How to read this report**

Highlight indicates a change.
Deleted indicates deleted content.
▲ indicates pages were changed.
⟷ indicates pages were moved.

[revised manuscript text omitted]